# The Antiviral Effect of Ephedrine Alkaloids-Free Ephedra Herb Extract, EFE, on Murine Coronavirus Growth in the Lung and Liver of Infected Mice

**DOI:** 10.3390/microorganisms13040830

**Published:** 2025-04-06

**Authors:** Akinori Nishi, Sumiko Hyuga, Masashi Hyuga, Masashi Uema, Nahoko Uchiyama, Hiroshi Odaguchi, Yukihiro Goda

**Affiliations:** 1TSUMURA Advanced Technology Research Laboratories, TSUMURA & CO., 3586 Yoshiwara, Ami-machi, Inashiki-gun, Ibaraki 300-1192, Japan; 2Oriental Medicine Research Center, School of Pharmacy, Kitasato University, 5-9-1 Shirokane, Minato-ku, Tokyo 108-8641, Japan; 3National Institute of Health Sciences, 3-25-26 Tonomachi, Kawasaki-ku, Kawasaki 210-9501, Japan

**Keywords:** murine hepatitis virus-infected mouse model, hepatitis, pneumonia, severe acute respiratory syndrome-like symptoms

## Abstract

Ephedrine alkaloids-free Ephedra Herb extract (EFE) was developed to reduce the adverse effects of Ephedra Herb, a constituent drug in Kampo medicines. It is produced by decocting Ephedra Herb with hot water and excluding the ephedrine alkaloids. EFE has analgesic and anti-cancer effects and inhibits respiratory viruses in vitro. To assess the pharmacological action of EFE in vivo, we evaluated its effect on the replication of murine hepatitis virus (MHV), a coronavirus that causes hepatitis, pneumonia, and severe acute respiratory syndrome-like symptoms, within infected mice. On Day 0, MHV was inoculated intranasally into female BALB/C mice, and EFE was orally administered once/day at 350–700 mg/kg (n = 10/group) starting 1 h after inoculation until Day 5. Through a plaque assay, MHV was detected on Day 5 in the lung and liver in all inoculated mice, but the titer was significantly lower in the EFE groups as compared with untreated control mice. Although not statistically significant, the clinical score for respiratory irregularity tended to be lower in the EFE treatment groups. In conclusion, EFE inhibits MHV replication in an in vivo mouse model of human coronavirus infection and exerts pharmacological action in the lung and liver.

## 1. Introduction

In Japan, the traditional herbal medicine Kampo has been employed extensively as an important therapeutic modality in combination with western drugs for various clinical disorders [1,2,3]. Kampo medicine originated from ancient Chinese medicine and was introduced to Japan in the 5th to 6th centuries; since then, diagnosis and treatment methods utilizing Kampo medicine have developed independently in accordance with practices suitable for Japan. As a result, Kampo medicine is defined by the World Health Organization (WHO) as ‘the medicine traditionally practiced in Japan, based on ancient Chinese medicine’ to distinguish it from ‘traditional Chinese medicine’ [1,2,3]. Over the years, Kampo medicine has been developed and incorporated within modern treatment practices; currently, 148 prescriptions for Kampo medicine are approved by the Ministry of Health, Labour and Welfare and are covered by national health insurance programs in Japan [1,2,3]. To ensure the quality of Kampo medicines and the constituent medicinal herbs, detailed characteristics are recorded in the Japanese pharmacopeia; in addition, based on regulations, Kampo medicines are manufactured and supplied in accord with both Japanese good manufacturing practice and quality control [1,2].

Against this background, many kinds of Kampo medicines are widely prescribed in Japan to treat common diseases, such as the symptom for the common cold, menopausal disorders, and fatigue based on patient indications; moreover, recent evidence supports the mechanism of action of Kampo medicine as a multi-compound drug, and Kampo medicine is being used to treat modern multifactorial diseases [1,2,3,4,5]. For instance, Daikenchuto, which is prescribed to relieve abdominal coldness and pain accompanied by abdominal flatulence, acts on the intraluminal transient receptor potential ankyrin 1 (TRPA1) activator and mediates the release of adrenomedullin and serotonin [1,6], leading to physiological responses such as vasodilation, motility, secretion, and pain signaling. Daikenchuto is now prescribed for gastrointestinal disorders including postoperative ileus and for alleviating the side effects of anti-cancer drug therapy [1,6], and its health economic benefits have been demonstrated [7]. Rikkunshito, another Kampo medicine, releases the orexigenic peptide ghrelin and has ameliorative effects on functional dyspepsia [8,9,10]. In terms of psychiatric problems, Yokukansan is currently used as a remedy for behavioral and psychological symptoms of dementia (BPSD), such as aggressiveness, agitation, and hallucinations, due to evidence of its effects on serotonergic and glutamatergic signaling [11,12,13,14,15]. Lastly, there is accumulating evidence supporting the efficacy of Kampo medicines prescribed for acute respiratory infection such as influenza virus infection and respiratory syncytial virus (RSV) infection, such as Maoto [16,17,18,19,20,21,22]. Therefore, it is expected that many kinds of medicinal herbs including Kampo medicines may have potential beneficial effects on disease caused by severe acute respiratory syndrome coronavirus 2 (SARS-CoV-2).

In 2023, the number of SARS-CoV-2 cases exceeded 760,000,000 worldwide with 6,000,000 deaths [23]. Since the start of the pandemic in 2019, many drugs and vaccines targeting SARS-CoV-2 have been developed and widely used to prevent infection and reduce symptoms. However, the virus continues to evolve, with genetic variance associated with a reduction in vaccine and antivirus drug efficacy; furthermore, patients with underlying medical conditions remain at a higher risk for severe disease symptoms and require various treatments for infection. Moreover, in addition to acute symptoms of the disease, some patients are affected by symptoms that remain long after the infection, known as long COVID, including the activation of the immune response, endothelial dysfunction, cognitive impairment, brain fog, and chronic fatigue [24]. Therefore, it is necessary to develop different approaches for the treatment and therapy of COVID-19.

In this regard, clinical studies of Kampo medicines have been conducted [25,26]. One randomized control study tested the efficacy of a Kampo medicine developed to treat the Spanish flu from 100 years ago, comprising Kakkonto, which is used to treat the common cold, coryza, febrile diseases, inflammatory diseases, neuralgia in the upper body, and urticaria, and Shosaikotokakikyosekko, which is used to relieve painful swollen throats or tonsillitis. As compared with standard antipyretics and antitussives, Kakkonto with Shosaikotokakikyosekko did not improve the relief of fever, cough, sputum, fatigue, or shortness of breath; however, recovery was significantly faster in the group treated with the Kampo medicine than in the control group, raising the possibility that traditional phytomedicines might have potential in counteracting coronavirus pandemics. In this context, we are interested in the antiviral activity of Ephedra Herb, a constituent of Maoto, which is prescribed for the common cold and has beneficial effects in the initial phase of influenza infection. The administration of Maoto decreases viral titers and exerts an antipyretic effect, as well as ameliorates virus-induced pneumonia. Maoto and Ephedra Herb also have activity against respiratory syncytial virus (RSV) in vivo and in vitro [16,17,21]. Maoto clinically improves flu symptoms with efficacy comparable to that of neuraminidase inhibitors in adults infected with the influenza A virus [19] and is effective as a post-exposure prophylaxis against COVID-19. Collectively, therefore, the evidence suggests that ephedra herb is a key ingredient of Kampo medicine that plays a role in its various antiviral activities [21,27,28].

Although Ephedra Herb is a primary constituent in the Kampo formulae, its ingredients, ephedrine alkaloids, have been associated with adverse effects, including acute myocardial infarction, hypertension, myocarditis, tachycardia, and lethal cardiac arrhythmia [29,30]. These effects are caused by the activation of the sympathetic nerve and the stimulation of α- and β-adrenergic receptors in several organs (e.g., heart, blood vessels, and bladder), raising major concerns about its use [31,32,33,34,35]. We therefore developed the Ephedrine alkaloids-free Ephedra Herb extract (EFE)—a hot water extract of Ephedra Herb originating from *Ephedra sinica* in which the ephedrine alkaloids have been removed by cation exchange resin [36]. EFE has various pharmacological actions, including anti-influenza virus activity, anti-RSV activity, anti-cancer activity, and analgesic properties, with reduced adverse effects relative to those of ephedrine alkaloids [37,38,39,40,41,42,43]. For instance, Huang et al. evaluated the effects of EFE on the anti-cancer drug, paclitaxel; it induced peripheral neuropathic pain in a mouse model and revealed that the oral administration of EFE (700 mg/kg) for 5 days significantly showed both preventive and therapeutic effects against peripheral neuropathic pain. Nakamori et al. and Hyuga et al. also indicated that the administration of EFE (700 mg/kg) significantly ameliorates formalin-induced pain [37,38,39]. These results clearly show that although EFE lacks ephedrine, a known active ingredient in Kampo medicine, it still contains potentially active components, and EFE has significant ameliorative effects in vivo. In addition, regarding EFE and its active ingredient, we reported that Ephedra Herb macromolecule condensed tannin (EMCT) has antiviral effects on SARS-CoV-2 in vitro [44]. Collectively, these observations have important implications for the clinical use of EFE against broad viral infections, including SARS-CoV-2, with few adverse effects. Furthermore, we conducted a double-blind randomized control clinical trial to evaluate the safety and efficacy of EFE on mild COVID-19, which showed that EFE caused no safety issues in COVID-19 patients, and although there were no differences in the non-aggravation rate of symptoms between the placebo and EFE groups, EFE treatment reduced nasal symptoms [45]. Thus, EFE has potential for the treatment of respiratory tract infections such as COVID-19 and influenza.

Therefore, to evaluate the efficacy of EFE against respiratory tract infections in vivo, we have explored its effects on infection caused by a murine coronavirus, murine hepatitis virus (MHV), in mice. It is reported that, after the infection of MHV, which belongs to the order Nidovirus in the family Coronaviridae, the virus multiplies in the liver and lung, resulting in symptoms similar to SARS-CoV-2 infections such as hepatitis and pneumonia; therefore, this model is considered suitable for estimating the influence of EFE on SARS-CoV-2 infections in humans [46,47,48,49].

## 2. Materials and Methods

### 2.1. Materials

EFE was prepared in accordance with our previous studies [36,37,44]. In brief, Japanese pharmacopeia-grade Ephedra Herb (*E. sinica*) was extracted in water at 95 °C for 1 h and then filtered, and the residue was washed with water. The filtered extract was centrifuged at 1800× *g* for 10 min, and the supernatant was passed directly through a DIAION SK-1B ion-exchange resin. The unabsorbed fraction was adjusted to a pH of 5 using 5% NaHCO_3_ (aq.) and evaporated under reduced pressure to obtain EFE, which was stored at -80 °C until use. For the in vivo experiment, EFE was dissolved in distilled water immediately before oral administration.

SR-CDF1 DBT cells (JCRB1580), a mouse cell line derived from a murine brain tumor, were obtained from the Japanese Collection of Research Bioresources Cell Bank in the National Institutes of Biomedical Innovation, Health and Nutrition. Antibiotic-Antimycotic and minimum essential media (MEM) were obtained from Thermo Fisher Scientific (Waltham, MA, USA); bovine serum albumin (BSA) and sodium hydrogen carbonate (NaHCO_3_) from FUJIFILIM Wako Pure Chemical Corporation (Richmond, VA, USA); D-glucose and L-glutamine from Tokyo Chemical Industry; DEAE dextran from pK chemicals A/S; Agar Noble from Becton, Dickinson, and Company (Franklin Lakes, NJ, USA); and Hank’s balanced salt solution (HBSS) from Life Technologies Corporation (Carlsbad, CA, USA).

### 2.2. Viral Culture

Murine hepatitis virus (MHV-1, ATCC VR-261) was originally obtained from ATCC and stocked at Nihon Bioresearch Inc. (Gifu, Japan). MHV was cultured in SR-CDF1 DBT (JCRB1580 cells) obtained from stocks at Nihon Bioresearch Inc. The culture medium was collected for mouse inoculation.

### 2.3. Animals

The study was approved by the Laboratory Animal Committee of TSUMURA & CO. (Tokyo, Japan) and Nihon Bioresearch Inc (Hashima, Japan). All animal experiments followed the guidelines of the Ministry of Health, Labour and Welfare, Japan.

In accordance with previous studies using animal models of the influenza virus, respiratory syncytial virus, and MHV infection, only female mice were used in this study [17,21,47,50]. Based on the previous study for susceptivity against the MHV infection and the disease phenotype [47], we decided to use the BALB/c Cr Slc (BALB/c) mice for this in vivo infection study, and the mice were obtained from Japan SLC, Inc. (Shizuoka, Japan) at 5 weeks of age and habituated for 5 days. Throughout the experimental period, they were housed in cages lined with paper chips (five mice per cage) with nesting material (Animec) as enrichment and had ad libitum access to food (CRF-1, Oriental Yeast, Tokyo, Japan) and water. Relative humidity, temperature, and the 12 h light/dark cycle were maintained at 46–64%, 21–25 °C, and 6:00–18:00, respectively.

### 2.4. In Vivo MHV Inoculation and Animal Evaluation

The BALB/c mice were given an intranasal inoculation of MHV at a dose of 1 × 10^5^ plaque-forming units (PFU)/0.05 mL/mouse at 6 weeks of age [49] after the habituation period. The mice were divided into five treatment groups (n = 10 per group) as follows: (1) no-inoculation as a control for MHV inoculation (NI group); (2) MHV inoculation as a no-treatment control (MI group); (3) MHV with EFE treatment at (700 mg/kg × 2)/day [EFE_(700×2)_ group]; (4) MHV with EFE treatment at 700 mg/kg/day [EFE_(700)_ group]; and (5) MHV with EFE treatment at 350 mg/kg × 2/day [EFE_(350×2)_ group]. The dosage of EFE was taken from previous studies on the safety and analgesic effect of EFE [37,39]. The mean body weight of each group on the initial day of MHV infection (Day 0) was 16.18 ± 0.62 g (NI group), 16.15 ± 0.69 g (MI group), 16.21 ± 0.63 g (EFE_(700×2)_ group), 16.12 ± 0.74 g (EFE_(700)_ group), and 16.05 ± 0.62 g (EFE_(350×2)_ group); there were no statistical differences for body weight according to Bonferroni’s multiple comparison test (*p* > 0.05).

On Day 0, EFE (the EFE_(700×2)_ and EFE_(350×2)_ groups) or the distilled water control (the NI and MI groups) was administered 1 h after inoculation (8:00–11:00) and then 4 h later; EFE or water was then administered twice daily (8:00–11:00 and 4 h later) in these groups for the next 5 days (Day 5); thus, EFE or water was administered 11 times in these groups. For the EFE_(700)_ group, EFE was administered 1 h after inoculation (8:00–11:00) (Day 0) and then once daily (8:00–11:00) for the next 5 days (Day 5); thus, EFE was administered 6 times in this group. Clinical signs, body weight, and rectal temperature were recorded. Clinical signs, rated from normal (0) to severe (3), were recorded for the following categories: (1) eye, (2) fur, (3) behavior, and (4) physiological symptoms (i.e., hypothermia, emaciation, and respiratory failure). Rectal temperature was measured before EFE treatment via a rectal thermometer (Model BAT-12, Physiotemp Instruments Inc., Clifton, NJ, USA).

On Day 5, all mice were anesthetized with an isoflurane inhalation solution (Mylan EPD G.K., Minato, Japan) 6 h after the last dose of EFE or water and sacrificed. After sacrifice, lung and liver tissues were collected and shredded into small pieces, homogenized in 2 mL of HBSS, and stored at −80 °C as the tissue homogenate for subsequent viral titer analysis.

### 2.5. Murine Hepatitis Viral Titer

The titer of replicated MHV was evaluated in the right lung and liver via a plaque assay. In brief, 0.1 mL of tissue homogenate, either undiluted or diluted 10- or 100-fold in MEM, was added to a 12-well cell culture plate containing SR-CDF1 DBT cells and incubated for 1 h. The cells were overlaid with 1.5 mL of an equal mixture of the culture medium (10×MEM, 10 mL; 7.5% NaHCO_3_, 3 mL; 200 mmol/L l-glutamine, 2 mL; 1% DEAE dextran, 1 mL; 15% glucose, 1 mL; 10% BSA, 1 mL; Antibiotic-Antimycotic, 1 mL; distilled water, 31 mL) and an agarose solution (0.8 g of agarose in 50 mL of distilled water). After culture for 1 day under 5% CO_2_ and 37 °C, the cells were overlaid with a medium containing neutral red and cultured for another day before viral plaques were counted.

### 2.6. Statistical Analysis

The measured values for rectal temperature, body weight, and viral titer were reported as the mean ± standard deviation (SD), and clinical sign scores were reported as the median with the interquartile range (IQR). Differences in values among groups were evaluated either by one-way analysis of variance (ANOVA) with Bonfferoni’s multiple comparisons test (viral titer, body weight, and rectal temperature) or by the Kruskal–Wallis test with Dunn’s multiple comparisons test (clinical sign scores) to compare the score between MI and NI for MI, each EFE group, and among EFE groups. GraphPad Prism 10 was used for data analysis (GraphPad, San Diego, CA, USA). The level of statistical significance was set at a *p* value of 0.05, and the results of the statistical test in each experiment were described at a level of *p* < 0.01 or *p* < 0.05.

## 3. Results

The present study evaluated the effect of EFE administration at three different doses on MHV infection in mice. One mouse in the MI control group, which was judged to have reached the humane endpoint for rectal temperature at Day 5 after MHV inoculation, was sacrificed early and not included in the following analysis. During the experiment, the EFE groups did not present the phenotype associated with safety, and there was no concern about the adverse event after the administration of EFE for the infection.

Figure 1 shows the viral titers of MHV recovered from mouse lung and liver tissues. MHV was recovered from both tissues on Day 5 after inoculation in the MI group (lung: 1622.3 ± 815.7 × 10^2^ PFU/g; liver: 1936.9 ± 958.3 × 10^2^ PFU/g) (Figure 1a,b). As compared with the MI group, the recovered viral titers were significantly decreased in all three EFE treatment groups (*p* < 0.001 in each group) for both lung [EFE_(700×2)_, 96.1 ± 33.8 ×10^2^ PFU/g; EFE_(700)_, 250.1 ± 214.6 × 10^2^ PFU/g; EFE_(350×2)_, 417.5 ± 384.2 × 10^2^ PFU/g] and liver tissues [EFE_(700×2)_, 66.4 ± 57.1 × 10^2^ PFU; EFE_(700)_, 159.8 ± 221.6 × 10^2^ PFU/g; EFE_(350×2)_, 269.7 ± 372.6 × 10^2^ PFU/g)] (Figure 1a,b). When we focused on the efficacy of each EFE group, each mouse of EFE_(700×2)_ tended to clearly decreased the virus titer’s relatively low variation. When we compared the effect of EFE_(700)_ and EFE_(350×2)_, the variety of EFE_(700)_ seemed to have less variation for the effect, with no significantly difference. The tendency of the effect was consistent against the infection in the lung and liver tissues.

Regarding the clinical sign scores, the MI group showed respiratory irregularity relative to the NI group (*p* < 0.01). Although the differences between the MI and EFE treatment groups were not significant, the score for respiratory irregularity tended to be lower in the EFE treatment groups (Figure 1c). The scores for three EFE groups did not differ and showed the same trend.

In terms of fur, the MI group showed a poor coat as compared with the NI group; however, the clinical score for coat was not improved by EFE treatment [NI, 0; MI, 2(2-2); EFE_(700×2)_, 2(2-2); EFE_(700)_, 1(1-2); EFE_(350×2)_, 1(1-2)]. After MHV inoculation, the mice showed no clinical signs of poor eye health or behavior as compared with the NI group; the scores in all five treatment groups were zero for these two categories.

Table 1 summarizes the effect of MHV inoculation and EFE treatment on body weight and rectal temperature on Day 5. Body weights differed significantly among the groups according to one-way ANOVA (F_(4,45)_ = 13.53, *p* < 0.01). The body weight of the MI group was significantly lower than that of the NI group (*p* < 0.01). The mean body weight of each EFE treatment group was significantly lower than that of the NI group (all *p* < 0.01), and there were no differences between the MI and EFE treatment groups. The scores of body weight in three EFE-administered groups presented the same trend, and there were no differences between the concentration and timing of administration. Rectal temperatures differed significantly among the groups according to one-way ANOVA (F_(4,44)_ = 4.916, *p* < 0.01). It significantly different between the MI and NI groups, and there were no differences between the MI and EFE treatment groups. The rectal temperature in the EFE_(700×2)_, EFE_(700)_, and EFE_(350×2)_ groups was not significantly different among groups, and the differences were within the range of normal daily alteration observed in the NI group and not notable.

## 4. Discussion

Our previous studies clearly revealed the antiviral effects of EFE treatment on different viral infections, including SARS-CoV-2, influenza, and RSV in vitro, as well as its analgesic action in an in vivo mouse model [37,38,42,44]; however, the ameliorative effect of EFE on infectious diseases in vivo remained to be investigated, and it provide the important evidence of EFE on COVID-19. Albuquerque et al. previously reported that MHV-1 infection of BALB/c mice, which is known as highly susceptible for several kinds of MHV strains, results in a SARS-like phenotype with histopathology in lung tissue such as interstitial pulmonary infiltrates [47]; therefore, in this study, we evaluated the efficacy of EFE treatment on MHV infection in BALB/c mice, with EFE administered at a dose (700 mg/kg) that showed analgesic action in vivo [37]. In addition, to gain insights into the relationship between the number of administrations and the stability of efficacy, we compared doses given once and twice daily at equivalent concentrations. Such data will be essential for the clinical application of EFE.

In this study, we found that the administration of EFE significantly decreased MHV titers in the lung and liver tissues of mice infected with MHV, with a tendency toward the amelioration of respiratory symptoms. In addition, we observed that the viral titer was lower in the group that received a single 700 mg/kg dose than in the group that received a 350 mg/kg dose twice daily. These results suggest that raising the blood concentration of the active components of EFE at a single timepoint is important to stably inhibit MHV replication. In other words, increasing the concentration of the active ingredients by administering a higher dose of EFE in the bloodstream is important for its antiviral effect.

In human, β-coronaviruses cause acute respiratory infections including the common cold, Middle East respiratory syndrome, and COVID-19. Similarly, MHV causes severe infectious diseases such as hepatitis and pneumonia in mice [46,47,48,49]. Both viruses enter cells via the endocytic pathway: in human, SARS-CoV-2 binds to the angiotensin-converting enzyme 2 (ACE2) receptor; in mouse, MHV binds to the carcinoembryonic antigen-related cell adhesion molecule 1 (CEACAM1) receptor [46,47,48,51,52]. Based on these observations and the fact that MHV-1 infection of BALB/c mice results in a SARS-like phenotype [47], we used these mice as a relevant disease model for human COVID-19. Our results show that EFE clearly decreased MHV replication within mouse organs after intranasal infection. This suggest that it may be an effective therapy against various upper respiratory viral infections, including SARS-CoV-2. As the next step, we need to evaluate in detail the pathophysiology of MHV infection, the active ingredients of EFE, and the mechanism underlying the action of EFE.

As a limitation of this study, we administered the EFE one hour after MHV inoculation because we obtained the ameliorative effect of early phase EFE of the infection; however, for the next study, it is worth knowing the effective interval between infection and administration to maintain the antivirus activity to provide the evidence for the clinical situation. In addition, we need to compare the efficacy of EFE and other anit-SARS-CoV-2 treatments such as antivirus small molecule treatment, remdesivir, and anti-spike protein mono-clonal anti-body treatment [53]. The study is important not only to know the relative potency of EFE against the other therapeutics but also to reveal the potential of combining EFE and other drugs, and this will lead to the possibility of treating COVID-19 and combating drug resistance. Furthermore, while we clearly revealed that EFE has anti-MHV effects, one basic question is whether the efficacy of EFE is equivalent to that of Ephedra Herb, which is similar to our previous findings on analgesic action. We will address this issue in a future study. Furthermore, it remains necessary to comprehensively clarify the candidate medicinal ingredients contained in EFE and their activity and pharmacokinetics. We did not investigate the active ingredients of EFE against virus infection. Besides ephedrine alkaloids, various candidate active ingredients such as flavonoids, herbacetin, tannins, and organic acids have been identified [54,55]. Among them, herbacetin, which is an aglycon of herbacein glycoside, was found to significantly suppress the hepatocyte growth factor-induced motility of human breast cancer MDA-MB-231 cells by inhibiting c-Met and Akt phosphorylation, while ephedrine alkaloids did not affect this signaling pathway [56]. In addition, high-molecular-mass condensed tannins with a molecular weight of 45,000 to 100,000, designated EMCT, are present in approximately 20% of EFE and have been shown to have analgesic, anti-cancer, and anti-influenza effects [57]. We previously showed that EFE affects several variant strains of SARS-CoV-2 in vitro, and EMCT is a key ingredient in this antiviral activity [44]; therefore, EMCT is likely to be a primary candidate for the anti-MHV activity of EFE observed in this study. Furthermore, the active ingredients in EFE have effects not only for MHV, variants of SARS-CoV-2, and seasonal influenza but also have a broad anti-virus spectrum including inhibitory effects for pandemic influenza and Middle East respiratory syndrome, and the accumulation of knowledge is important to reveal the specific treats and critical potential of EFE.

Lastly, the mechanisms underlying the antiviral effects of EFE on MHV replication in vivo, such as whether EFE prevents viral entry into the cell or inhibits viral proliferation by affecting the synthesis of viral proteins within infected cells, remains to be elucidated in future investigations. Furthermore, it is now well known that the acute symptoms of COVID-19 such as fever, acute respiratory symptoms, severe pneumonia, the chronic symptoms known as long COVID, which causes multiple symptoms; the activation of the immune response; endothelial dysfunction; cognitive impairment; brain fog; and chronic fatigue [24] are important clinical problems and require appropriate treatment. However, the comprehensive treatment of COVID-19 including chronic symptoms is now insufficient. Our previous studies significantly revealed the multiple efficacies of EFE on many kinds of diseases including anti-virus infection, anti-cancer activity, anti-cancer drug-induced peripheral neuropathic pain, and evidence for the multiple function of EFE shows us the therapeutic potential of EFE against the complex symptom for COVID-19.

## 5. Conclusions

The administration of EFE clearly decreased MHV viral titers in vivo and exerted pharmacological action in the lung and liver, with a tendency to decrease the clinical score for respiratory irregularity. Hyuga et al. previously showed that EFE has anti-influenza activity equivalent to that of Ephedra Herb in vitro [37], and Uema et al. revealed that EFE and EMCT significantly inhibit SARS-CoV-2 replication in vitro [44]. Furthermore, we revealed the certain potential of EFE for symptom relief against COVID-19 through a double-blind randomized control clinical trial [45]. Although we did not evaluate the ingredients that were active against MHV in EFE, we cannot rule out the possibility that ephedrine alkaloids in Ephedra Herb also have anti-MHV activity; our results strongly suggest that highly potent factors associated with antiviral effects are present in EFE. Furthermore, EFE is not associated with the typical adverse effects (e.g., excitation, insomnia, and arrhythmias) caused by ephedrine alkaloids in Ephedra Herb [41]. In the future, it expected that EFE will be useful in treating coronavirus infection with few serious adverse effects.

## 6. Patents

Kitasato University, Matsuyama University, National Institute of Health Sciences, and Tokiwa Phytochemical Co., Ltd. and TSUMURA & CO. are applicants for the relevant patent.

## Figures and Tables

**Figure 1 microorganisms-13-00830-f001:**
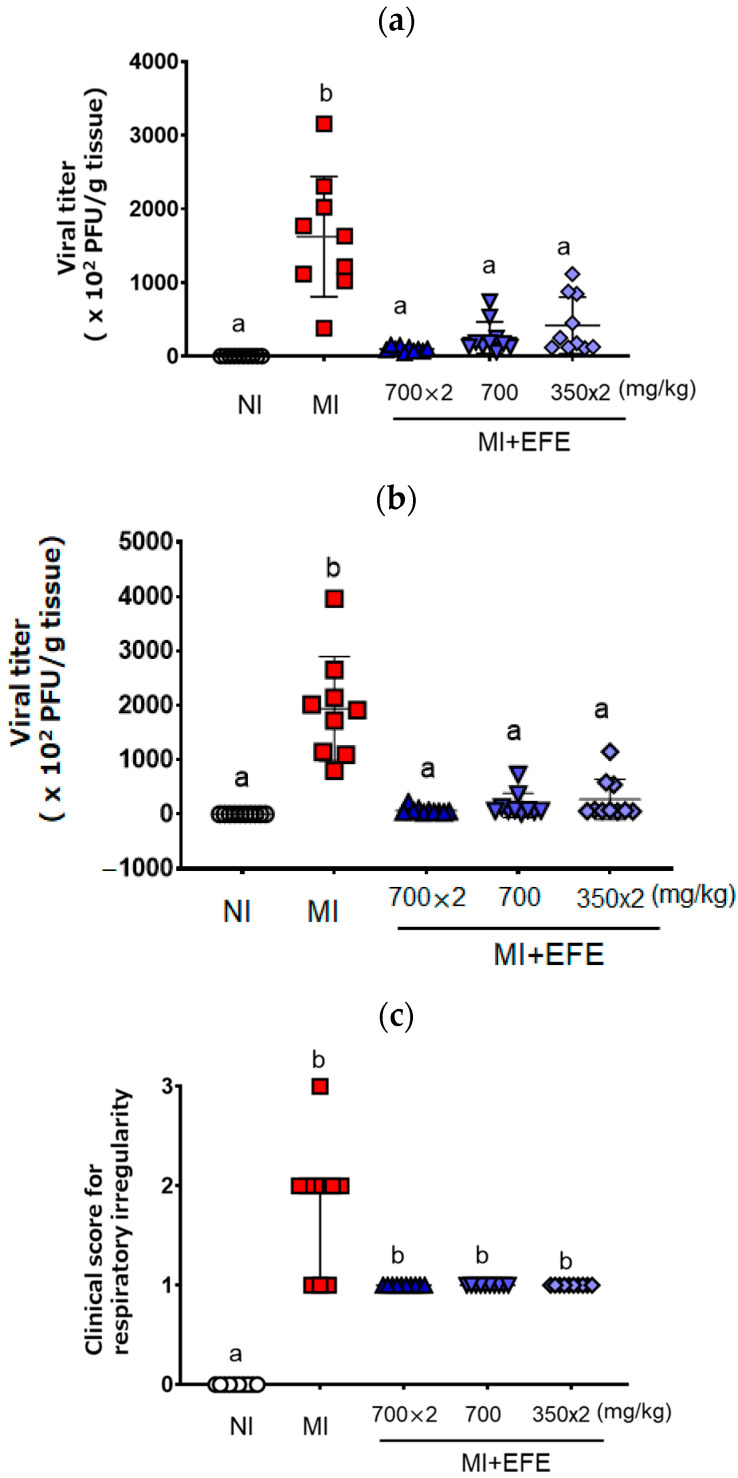
Viral titers recovered from the lung (**a**) and liver (**b**) and clinical score for the respiratory irregularity (**c**) of MHV-infected mice with or without EFE treatment. Viral titer (×10^2^ plaque-forming unit (PFU)/g tissue) is presented as both individual scores and mean ± SD (n = 10 for NI and EFE groups; n = 9 for MI), and the clinical score is presented as both individual scores and the median with IQR (n = 10 for all groups). A *p* value of < 0.01 via the one-way ANOVA with Bonferroni’s multiple comparisons test is seen for (**a**) and (**b**), and *p <* 0.01 via the Kruskal–Wallis test with Dunn’s multiple comparison test is seen for (**c**). Different letters indicate a significant difference among groups.

**Table 1 microorganisms-13-00830-t001:** Effect of MHV inoculation and EFE treatment on body weight and rectal temperature on Day 5.

	No Inoculation	Inoculation	EFE		
			700 × 2 mg/kg	700 mg/kg	350 × 2 mg/kg
Body weight	16.6 ± 0.9 a	14.3 ± 1.1 b	14.1 ± 0.9 b	14.1 ± 1.0 b	14.1 ± 0.9 b
Rectal temperature	36.5 ± 0.3 a	37.2 ± 0.6 a	37.0 ± 1.0 a	37.4 ± 0.3 a	37.5 ± 0.2 a

Data are mean ± SD (rectal temperature: n = 10 for NI and each EFE group; n = 9 for the MHV inoculation group; body weight: n = 10 for all groups). A *p* value of < 0.01 using one-way ANOVA with Bonferroni’s multiple comparisons test indicates significance. Different letters indicate a significant difference among groups.

## Data Availability

The data in this study could be inquired from the corresponding author.

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
