# Peer review of "The Antiviral Effect of Ephedrine Alkaloids-Free Ephedra Herb Extract, EFE, on Murine Coronavirus Growth in the Lung and Liver of Infected Mice"

_microorganisms, 2025, doi:10.3390/microorganisms13040830_

Round 1

Reviewer 1 Report

Comments and Suggestions for Authors

The article entitled "The Antiviral Effect of Ephedrine Alkaloids-Free Ephedra Herb Extract (EFE) on the Growth of Murine Coronavirus in the Lung and Liver of Infected Mice" has been evaluated.

However, before the manuscript can be considered for publication, substantial revisions are required in several critical areas:

This segment of the research has been previously published in the journal Microorganisms (2023, volume 11, article 298), titled "Antiviral Effect of Ephedrine Alkaloids-Free Ephedra Herb Extract against SARS-CoV-2 In Vitro" (doi:10.3390/microorganisms11020534). I didn’t find any significant advancement of the study other than the in-vvio study.

However, the study did not adequately characterize the chemical composition of the Ephedra Herb Extract, especially regarding the key constituents responsible for inhibiting Murine Coronavirus. It is imperative that a thorough chemical characterization is conducted using Infrared (IR) spectroscopy, Mass spectrometry, and Nuclear Magnetic Resonance (NMR) spectral analysis.

Furthermore, while the authors compared the inhibitory effects of the Ephedra Herb Extract against the Shyam control, it is essential to include a positive control in the experimental design. This addition would allow for a clearer understanding of the extract's actual efficacy in suppressing viral growth.

The rationale behind administering three different concentrations of Murine Hepatitis Virus (MHV) with the Ephedra Herb Extract treatment is unclear. Specifically, the treatment groups consisted of MHV with EFE at doses of (700 mg/kg × 2) daily [designated as EFE(700×2) group], MHV with EFE at a daily dose of 700 mg/kg [the EFE(700) group], and MHV with EFE at (350 mg/kg × 2) daily [the EFE(350×2) group]. Further clarification is needed to justify the selection of these specific dosages.

Additionally, the manuscript does not address the safety profile of the injected doses used in the study. It is crucial to establish the safety of these dosages for animal studies to ensure ethical compliance.

Lastly, the manuscript lacks a clear conclusion summarizing the findings of the research, which is vital to encapsulate the significance and implications of the study’s outcomes.

Comments on the Quality of English Language

 The English could be improved to more clearly express the research.

Reviewer 2 Report

Comments and Suggestions for Authors

This study introduces a novel Ephedrine alkaloids-free Ephedra Herb extract, developed to minimise the adverse effects of Ephedra Herb while retaining its pharmacological benefits, such as inhibiting murine hepatitis virus replication in a mouse model. Despite its promising findings, including reduced viral titers in the lung and liver, the study's limited dataset diminishes its strength and breadth. Consequently, the scope of the investigation aligns more closely with a short communication rather than a full original article.

  1. Ensure to incorporate keywords that differ from those used in the title.
  2. Some sentences must be supported by references. For example, lines 40-43 and so on.
  3. Please avoid one or two-sentence paragraphs. Expand the idea or connect paragraphs with similar ideas.
  4. Please do not include p-value in the abstract section.
  5. Why only female animals was used in this research?
  6. Please provide the weight of animals used in the in vivo experiments.
  7. How were the doses and the regimen treatment determined? Was it based on previous reports? How about the pharmacokinetics?
  8. Is not one hour after inoculation a short period? When do usually patients are diagnosed and start the treatment?
  9. Were the signs monitored in an unbiased way? Please clarify it. Did the authors knew the treatment and respective cages?
  10. Line 100. How were the tissues collected? In PFA, PBS or another buffer?
  11. Line 115. Why do the authors use SE and not SD? I consider SD more appropriate to visualise the deviation and variability in the experiments.
  12. The authors must revise the post-test used as they are establishing a comparison with a control group.
  13. How was the group size for the animals determined? Did the authors perform power calculations to justify their choice? The number appears excessively high and seems difficult to justify based on many of the methods I am familiar with. I am curious and really interested in the experimental design.
  14. Figures 1 and 2 can be combined in one figure with a, b and c.
  15. The quality of Table 1 must be improved.
  16. The figure legends are inconsistent with the methodology described. The authors mentioned using different post hoc tests, but this is not clearly reflected.
  17. Figures do not need to be again mentioned in the discussion section.
  18. The references mut be standardised. Sometimes words start with capital letter, others not.

Reviewer 3 Report

Comments and Suggestions for Authors

This study investigates the antiviral effects of Ephedrine alkaloids-free Ephedra Herb extract (EFE) against murine hepatitis virus (MHV) in a mouse model. The authors provide compelling in vivo evidence that EFE reduces viral titers in the lung and liver, suggesting potential applications in coronavirus treatment. I do have the following comments:

  1. While the study tested 3 different EFE doses, I think a discussion paragraph on the need for a pharmacokinetic/ pharmacodynamic study for optimal concentration and viral clearance over time
  2. While potential uses have been described, another discussion point could be on comparison with existing drugs such as remdesivir, etc.
  3. Kruskal-Wallis testing was done, which means that data should be median and IQR not mean and SE, as the test is non-parametric. Please address this in methods and results.
Comments on the Quality of English Language

Minor English corrections:

  1. "We need to carefully evaluate the pathophysiology of MHV and elucidate the mechanism of action of EFE." sound too informal;
  2. use "administer" instead of "given orally";
  3. missing word: "it is produced by decocting Ephedra Herb";
  4. "EFE are absorbed into blood" should be "absorbed into the bloodstream"

Reviewer 4 Report

Comments and Suggestions for Authors

The authors aimed to investigate the antiviral properties of an Ephedrine alkaloids-free Ephedra Herb extract (EFE) in the MS microorganisms-3541701. The inhibitory activity against the Murine hepatitis virus was in vitro evaluated using 50 experimental mice divided into 2 control groups and 3 study groups.

Introduction

It is too short in its current form; not all readers are familiar with Kampo medicine and its traditional formulations. 

The authors are invited to briefly present some essential principles of Kampo medicine and differentiate them from other Traditional Asian Medicines. Then, they can enumerate the most used species in Kampo medicine and show the complete scientific name of Ephedra sp., the source of EFE.

The bioactive phytochemicals of Ephedra sp can be described, and then the therapeutic importance of EFE and its maintained constituents can be discussed. The authors could indicate the previously studied EFE's pharmacological properties.

Finally, the authors can show the aim of the present study, its novelty, and further applications.

Materials and Methods

Please indicate in Materials (line 65) all chemicals, reagents, culture media, etc., used in the present study and their provenance.

The EFE preparation is missing even though it is essential to this work. The provided reference was published nine years ago (2016). Therefore, the authors are invited to show the plant source (with GPS coordinates, if necessary), the harvesting time, the voucher specimen, and the preliminary processes  (drying, storage, etc). The EFE preparation should be briefly described, and the extraction yield and storage conditions should be mentioned. 

The quantitative determination of the EFE constituents is missing. Is EFE used in this study as a standardized plant extract with high-quality ingredients and a rigorous standardization process? Please specify.

Lines 70-71: Please indicate the type/origin of cells, not only their notation.

LInes 91-92: The authors wrote: "The first dose of EFE was given orally 1 hour after inoculation (Day 0); it was then administered once a day for the next 5 days." Please rectify this because you also have [EFE(700×2) group] and [EFE(350×2) group]. 

Results

Please respect the same order as described in the Materials and Methods.

Please place each figure immediately after its first mention in the MS text to help readers understand the original results better.

Discussion

Please motivate the study design and justify why it does not include the EFE phytochemicals' quantitative evaluation.

Please improve the discussion of the original results by comparing all test groups and dividing the results by their value distribution in each experimental group to reveal the differences between the used doses and the administration (once or twice daily).

The reviewer considers that the main limitation is the absence of identifying and quantifying the bioactive compounds. Therefore, the in vivo results do not have solid support; the discussions are evasive because the phytochemical screening is missing. To confirm their results with other studies, the authors did not discuss which compounds are responsible for the antiviral activity. They only mentioned  "the active component of EFE" with a rising concentration in blood (lines 182-183), active ingredients  (line 184), and EMCT (lines  52-53, 186, 187), without other details. 

In both sections, Introduction and Discussion, the authors show the advantages of eliminating the Ephedrine alkaloids, but they do not reveal the identified and quantified phytochemicals responsible for the extract's antiviral effects.

After all revisions, the Conclusions should be reformulated. 

Comments on the Quality of English Language

Moderate revision

Round 2

Reviewer 1 Report

Comments and Suggestions for Authors

The authors made significant revisions based on the reviewer comments; it can be acceptable for publication.

Reviewer 2 Report

Comments and Suggestions for Authors

The authors have addressed all the reviewers' comments comprehensively. However, I recommend resubmission as a short communication, as the findings do not meet the criteria for publication as an original article.